# Transforming Waste into Value: Eco-Friendly Synthesis of MOFs for Sustainable PFOA Remediation

Atef El Jery [1], Renzon Daniel Cosme Pecho [2], Meryelem Tania Churampi Arellano [3], Moutaz Aldrdery [1], Abubakr Elkhaleefa [1], Chongqing Wang [4,*], Saad Sh. Sammen [5,*] and Hussam H. Tizkam [6]

[1] Department of Chemical Engineering, College of Engineering, King Khalid University, Abha 61411, Saudi Arabia; ajery@kku.edu.sa (A.E.J.); malddrdery@kku.edu.sa (M.A.); amelkhalee@kku.edu.sa (A.E.)
[2] Department of Biochemistry, Universidad San Ignacio de Loyola (USIL), Lima 15001, Peru; rcosme@usil.edu.pe
[3] Department of Industrial Engineering, Universidad de Lima, Lima 15001, Peru; mchuramp@ulima.edu.pe
[4] School of Chemical Engineering, Zhengzhou University, Zhengzhou 450001, China
[5] Department of Civil Engineering, College of Engineering, University of Diyala, Baqubah 10047, Iraq
[6] Pharmacy Department, Al Safwa University College, Karbala 56001, Iraq; tizgam@alsafwa.edu.iq
* Correspondence: zilangwang@126.com (C.W.); saad123engineer@yahoo.com (S.S.S.)

**Abstract:** In response to the need for sustainable solutions to address perfluorooctanoic acid (PFOA) contamination, we have developed an eco-friendly approach for synthesizing two types of metal-organic frameworks (MOFs) using waste polyethylene terephthalate (PET) bottles via a one-pot microwave-assisted strategy. Our innovative method not only avoids the initial depolymerization of PET bottles but also promotes environmental conservation by recycling waste materials. The La-MOF and Zr-MOF materials exhibit remarkable surface areas of 76.90 and 293.50 $m^2/g$, respectively, with La-MOF demonstrating greater thermal stability than Zr-MOF. The maximum experimental PFOA adsorption for La-MOF and Zr-MOF was obtained at 310 and 290 mg/g, respectively. Both MOFs follow the Langmuir isotherm closely, with the adsorption of PFOA following a pseudo-2nd-order kinetic model. In packed-bed column tests, breakthrough positions of 174 and 150 min were observed for La-MOF and Zr-MOF, respectively, with corresponding bed volumes of 452 mL and 522 mL based on the PFOA limit of 0.07 µg/L in drinking water. These findings indicate that these MOFs can be used in industrial packed-bed columns to remove PFOA from contaminated water sources in an efficient and cost-effective manner. Importantly, the sorption performance of the fabricated MOFs for PFOA remained stable, decreasing by less than 10% over seven cycles. This study underscores the potential of recycled PET bottles and the one-pot microwave-assisted synthesis of MOFs as an effective and environmentally friendly solution for PFOA remediation. This innovative approach has several managerial implications, such as the use of waste materials as a feedstock, which can reduce the cost of production and minimize environmental impact by promoting recycling and repurposing, enhancing the reputation of companies operating in the chemical industry, and improving their sustainability metrics. By integrating sustainability principles and waste recycling, our approach offers promising avenues for addressing PFOA contamination while promoting resource efficiency and environmental conservation.

**Keywords:** adsorption; PFOA removal; metal organic framework (MOF); water treatment

## 1. Introduction

In recent times, the treatment of water pollution has garnered a prodigious amount of attention from the scientific community [1–3]. Notably, PFOA, a persistent organic pollutant of paramount concern, has captured the attention of experts. PFOA is extensively employed in the textile, industrial, and equipment domains as a surfactant, fire retardant, fluoroelastomer, and textile finish, among others [4,5]. However, it is ubiquitous

in surface water, municipal sewage, and drinking water, causing tremendous distress to the ecosystem and public health [6,7]. PFOA is renowned for its stability, rendering it resistant to natural processes, biodegradation, and traditional decontamination techniques employed in wastewater or water treatment plants [8,9]. PFOA's deleterious effects on the hematological, lymphatic, and renal systems, as well as its carcinogenic toxicity to humans and other living organisms, are well documented [10,11]. Therefore, it is crucial to develop a cost-effective and innovative methodology to combat PFOA contamination in the aquatic system [1].

Recently, numerous approaches have been explored to mitigate the detrimental effects of PFOA. These methods have included adsorption, electrochemical methods, sonochemical methods, direct photolysis, microwave-Fenton, and photocatalysis [12–17]. Notwithstanding the recent advances in mitigating PFOA, a member of the per- and polyfluoroalkyl substances (PFASs), the efficiency of these methods is heavily impacted by the unique physicochemical properties of PFOA [18,19]. PFASs, with their variable carbon chain lengths and the substitution of fluorine atoms for hydrogen atoms, demonstrate tremendous resistance to degradation, oleophobic as well as hydrophobic traits, and heat durability, all of which are granted by the strong carbon-fluorine (C-F) bond and the great level of fluorination [20,21]. Currently, biological treatments can only cleave the C-C bond with negligible effects on the C-F bonds [22]. Even though electrochemical oxidation has a higher oxidative capacity for perfluorinated carboxylic acids than general oxidation and biological processes, it generates hazardous by-products [23–25]. For example, when long-chain PFAS compounds such as PFOA are degraded using oxidation methods, they can be transformed into shorter-chain PFAS molecules such as perfluoropentanoic acid (PFPeA) and perfluorobutanoic acid (PFBA). These shorter-chain PFAS molecules exhibit higher solubility and stability in water compared to their long-chain counterparts. Consequently, the treatment of long-chain PFAS compounds using oxidative methods poses an additional challenge of effectively removing these newly formed short-chain PFAS molecules from the water or wastewater streams [26–28]. Additionally, alternative methods such as sonolysis and photolysis, which employ sound waves or light for the degradation of PFAS compounds, along with the utilization of fungal enzymes, are currently in the experimental or developmental stage. The practical feasibility of implementing these approaches on a large scale has not yet been fully demonstrated [29,30]. In contrast, filtering procedures and sorption are comprehensive techniques that transfer PFAS particles from one environment to another medium without dealing with degradation [31]. The filtration technologies employed for PFAS removal largely stem from existing filtration methods that were initially developed for addressing other contaminants. This implies that the feasibility of these approaches for large-scale applications has been established. However, the unique chemical characteristics of PFAS compounds and their behavior in water necessitate careful consideration when devising new separation techniques specifically tailored for PFAS treatment [29,32].

In response to the persistent environmental and health risks posed by perfluorooctanoic acid contamination in aqueous solutions, research on adsorption-based approaches has been developed, focusing on various adsorbent materials with high adsorption capacities, such as granular activated carbon, clay, functionalized synthetic fibers, and silica zeolites [33–36]. However, due to the limitations of conventional adsorbents in terms of their cost, selectivity, efficiency, and kinetics, innovative and advanced strategies have become necessary. Among these promising approaches is the use of MOFs, a novel family of porous compounds that have demonstrated exceptional capabilities in PFOA removal [37–39]. MOFs are characterized by their coordination interaction among metal clusters and organic ligands, allowing for tunable and adjustable structures with high surface areas, exceptional porosity, and effective active sites [40,41]. Consequently, MOFs have been widely recognized for their versatile and promising applications across various fields [42].

The eco-friendly production of MOFs using waste materials has garnered tremendous attention in light of the pressing need to address plastic contamination in recent decades [43,44]. Particularly, the utilization of low-cost organic materials using waste

plastics has the potential to significantly decrease the production cost of MOFs [43,44]. PET-based waste materials constitute a substantial resource in urban centers, with a global annual production volume of 300 million tons, and this figure is anticipated to increase [45]. Given that terephthalic acid (BDC; $C_8H_6O_4$) obtained from waste PET materials can serve as a pivotal chemical for the synthesis of MOFs, this strategy represents an alluring approach for the mitigation of waste PET-based materials [46]. Furthermore, this scenario enables the design of bespoke functional adsorbents with the requisite properties for the effective elimination of PFOA from contaminated aqueous systems [42].

To address a significant research gap in the field of efficient, cost-effective, and sustainable removal of PFOA from aqueous media, we focused on the application of MOFs synthesized from PET bottles. To the best of our knowledge and based on an extensive review of the existing literature, this approach has not been explored previously. Our aim was to develop a cost-effective and environmentally friendly method for eliminating PFOA from water by synthesizing two distinct types of MOFs, namely La-MOFs and Zr-MOFs. We utilized PET bottles as a readily available and abundant source of terephthalic acid (BDC), the organic linker used in MOF fabrication. This not only reduced production costs but also offered a sustainable solution for PET waste management. Our innovative approach involved a one-pot microwave-assisted synthesis strategy, which significantly expedited the synthesis process while maintaining the quality of the resulting MOF materials. By bypassing the primary depolymerization of PET bottles, we simplified the synthesis process and achieved efficient MOF production. Therefore, our work introduces a novel method that combines sustainable PET waste management, microwave-assisted synthesis, and the application of MOFs for PFOA removal, contributing to the advancement of research in this field and providing a sustainable solution to address the persistent pollutant issue in aquatic systems. The utilization of microwave heating enables rapid and homogeneous reactions, leading to a significant reduction in reaction time and energy consumption [47]. We thoroughly characterized the resultant MOFs through multiple advanced techniques, such as thermogravimetric analysis (TGA), X-ray photoelectron spectroscopy (XPS), Fourier transform infrared (FTIR), powder X-ray diffraction (PXRD), and Brunauer–Emmett–Teller (BET). To ascertain the efficiency of the synthesized MOFs in PFOA removal, we conducted adsorption isotherm and kinetic analyses under both column and batch conditions. The influence of the solution pH and MOF material regeneration were also investigated for practical applications. Our findings advance the prospect of using PET waste-derived MOFs as a feasible and sustainable solution for the removal of PFOA from aqueous environments.

## 2. Materials and Methods

### 2.1. Chemicals

The chemicals utilized in the current work include ethanol, N,N-dimethylformamide, hydrochloric acid (DMF), zirconium chloride ($ZrCl_4$), lanthanum chloride tetrahydrate ($LaCl_3 \cdot 4H_2O$), acetic acid ($CH_3COOH$), and sodium hydroxide (NaOH). The chemical materials were bought from Sigma–Aldrich and utilized without additional purification as they were analytical grade. Transparent PET bottles were gathered from local sources, and their lids were removed prior to washing. Subsequently, the bottles were manually cut into pieces approximately 2 mm in size using scissors.

### 2.2. MOFs Synthesis Methodology

The PET waste bottles underwent a rigorous washing process using deionized water and were subsequently subjected to drying for 5 h at 70 °C. To synthesize Zr-based MOF adsorbent, 4.0 g of PET chips, measuring around 2 mm × 2 mm, was combined with 0.8 mL of acetic acid, 4 mmol of $ZrCl_4$, and 100 mL of DMF in a 150-mL Teflon autoclave. The autoclave was then subjected to microwave radiation, and heated until the desired temperature of 160 °C was attained within 10 min. Following a holding period of 45 min, the autoclave was cooled down to 25 °C. The resultant precipitates were isolated via centrifugation at 7000 rpm and subjected to washing with a DMF and ethanol mixture

to remove residual impurities from the as-synthesized products. Eventually, the solid materials were subjected to a hot air-drying process for 15 h at 90 °C. Moreover, La-MOF was fabricated using the same methodology utilized for the synthesis of Zr-MOF, with $LaCl_3 \cdot 4H_2O$ replacing $ZrCl_4$.

### 2.3. Characterization Techniques

TGA was performed utilizing the PerkinElmer Pyris Diamond, a thermogravimetric analyzer. The temperature ramp ranged from ambient conditions to 800 °C, with a heating rate of 10 °C/min, all while maintaining a nitrogen gas flow. Throughout the experiment, the weight loss and thermal decomposition characteristics of Zr-MOF and La-MOF were tracked across varying temperatures. For nitrogen adsorption-desorption isotherm measurements, an automated gas adsorption system, the Micromeritics TriStar II Plus, was employed. The measurements were conducted at a temperature of −196 °C. Prior to the analysis, the samples underwent degassing for a duration of 4 h under vacuum conditions at 200 °C. As part of the analysis process, FTIR analysis was conducted at different stages on the Zr-MOF and La-MOF. Spectra in the range of 4000–400 $cm^{-1}$ were recorded utilizing the PerkinElmer Spectrum Two spectrometer, which possessed a resolution of 4 $cm^{-1}$. To prepare the FTIR samples, the materials were mixed with KBr and compressed into pellets. To determine the elemental composition and chemical state of the materials, XPS spectra were recorded using the Thermo Fisher Scientific K-Alpha+ spectrometer. The samples were excited with a monochromatic Al Kα X-ray source (hν = 1486.6 eV), and the emitted photoelectrons were meticulously analyzed to gather information regarding the surface chemistry of the materials. Lastly, XRD patterns were obtained using the Bruker D8 Advance diffractometer equipped with Cu Kα radiation. The instrument operated at 30 mA and 40 kV, scanning the 2θ range from 2° to 50° with a step size of 0.02°.

### 2.4. Experimental Methodology for PFOA Removal under Batch Conditions

The current section details the PFOA removal procedure using Zr- and La-MOF from aqueous solutions. To create the stock solution of PFOA, 105.26 mg of PFOA powder (95%) was dissolved in 1 L of water and thoroughly mixed until complete dissolution was achieved. This resulted in a stock solution with a concentration of 100 mg/L. The stock solution served as the starting point for further dilutions to generate sample solutions with concentrations ranging from 1 to 20 mg/L. In the PFOA removal procedure, 5.0 mg of the solid adsorbents were introduced to 100 mL of PFOA sample solution at concentrations ranging from 1 to 20 mg/L. To achieve adequate contact between the adsorbent and the PFOA molecules, the solution was then kept under agitation for 24 h at 25 °C. The resultant mixture was then filtered, and the concentration of PFOA in the filtrate was analyzed using Liquid Chromatography–Mass Spectrometry (LC–MS) analysis, which is explained in detail in Section 2.5. The PFOA adsorption capacity onto MOFs was obtained using the following equations:

$$\text{PFOA adsorption capacity} = q_e = (C_i - C_e) \times \frac{V}{m} \tag{1}$$

$$\text{PFOA adsorption percentage} = (C_i - C_f) \times \frac{100}{C_i} \tag{2}$$

Here, $C_i$ (mg $L^{-1}$) denotes the initial amount of PFOA, while $C_f$ (mg $L^{-1}$) attributes the PFOA amount after the treatment. V indicates the total volume of the samples (L), and m defines the weight of the adsorbent used for treatment in g.

### 2.5. Liquid Chromatography–Mass Spectrometry (LC–MS) Analysis

LC–MS analysis was conducted to enable comprehensive quantification of PFOA. The analysis was carried out using an LC–MS system comprising an Agilent 1260 Infinity II series high-performance liquid chromatography (HPLC) instrument coupled with an Agilent 6470 Triple Quadrupole Mass Spectrometer. This setup facilitated the precise

identification and quantification of PFOA. For chromatographic separation, an Agilent Zorbax Eclipse XDB-C18 column was utilized. The column provided efficient separation and resolution of PFOA. The LC mobile phase consisted of a solvent mixture of high-purity methanol and water, prepared using HPLC-grade solvents. This solvent system ensured optimal chromatographic performance and reproducibility. During the LC–MS analysis, PFOA was detected and quantified based on its specific mass-to-charge ratio (m/z) and retention time. Calibration curves were constructed using known concentrations of PFOA standards to accurately determine the concentration of PFOA in the samples.

### 2.6. Experimental Procedure for Isotherm and Kinetic Studies

The isotherm studies were performed using Freundlich and Langmuir isotherm models. The experimental data obtained from the PFOA adsorption process were fitted to these models. The corresponding equations are as follows:

$$\frac{1}{q_e} = \frac{1}{C_e \times q_m \times K_l} + \frac{1}{q_m} \tag{3}$$

$$Logq_e = LogK_f + \frac{1}{n} \times LogC_e \tag{4}$$

This equation involves several variables, including $q_m$ (mg/g), which represents the theoretical value for maximum PFOA adsorption, $C_e$ (mg/L) denotes the PFOA concentration at equilibrium, $K_l$ (L/mg) denotes the Langmuir constant, and $K_f$ (mg/g) and $n$ denote the constant parameters of the Freundlich model.

The kinetic study evaluated the influence of contact time between the adsorbent and PFOA particles on the removal process over a range of 5–30 min. The experimental conditions were an adsorbent dose of 5 mg, sample volume of 100 mL, and PFOA concentration of 20 mg/L, where different samples were prepared for different contact times. Experimental data obtained from the PFOA adsorption process were fitted to the pseudo-2nd and 1st-order kinetic models. The corresponding equations (Equations (5) and (6), respectively) are as follows:

$$Ln(q_m - q_t) = Lnq_m - K_1 \times t \tag{5}$$

$$\frac{t}{q_t} = \frac{1}{K_2 \times q_m^2} + \frac{t}{q_m} \tag{6}$$

This equation involves several variables including $q_m$, which represents the maximum amount of PFOA that can be adsorbed per unit of adsorbent mass, $q_t$, which denotes the amount of PFOA adsorbed at a specific time point, $K_1$ attributes the rate constant for the pseudo-1st-order kinetic model, $K_2$ refers to the rate constant for the pseudo-2nd-order kinetic equation, and $t$ denoting the duration of the adsorption process in minutes.

### 2.7. Experimental Procedure for pH Studies

The pH study was conducted using PFOA sample solutions with a concentration of 10 mg/L, an adsorbent amount of 5 mg, and a solution volume of 100 mL. The pH of the solution was adjusted from 2 to 9 using hydrochloric acid and sodium hydroxide solutions. The mixtures were stirred at 25 °C for 24 h to establish equilibrium. The concentration of PFOA in the filtrate was then analyzed using LC–MS analysis, and the corresponding adsorption pH curve was constructed.

### 2.8. Desorption Procedure of PFOA from the MOF Surface

To regenerate the PFOA-loaded Zr-MOF and La-MOF adsorbents and facilitate the desorption of PFOA from their surfaces, a regeneration procedure using NaOH was employed. The PFOA-loaded MOF samples were immersed in a 0.5 M NaOH solution, ensuring complete coverage of the MOF surfaces. Gentle stirring was applied for 24 h under controlled room temperature conditions. After the desorption process, the MOF samples were separated from the NaOH solution via centrifugation. A thorough rinsing with deionized water

was performed to remove any remaining NaOH and PFOA residues. The regenerated MOF samples were then oven-dried at 100 °C to remove excess water.

### 2.9. Experimental Methodology for PFOA Removal under Column Conditions

The fixed-bed column experiments were carried out at ambient temperature utilizing a 35 cm high column with an internal diameter of 2 cm, with a stainless-steel bottom that was covered via glass wool. The adsorbents were meticulously packed into the column to obtain a uniform bed height of 2 cm. The peristaltic pump was employed to precisely transfer the contaminated water, containing a PFOA concentration of 600 µg/L, into the column at a controlled flow rate of 5 mL/min. Subsequently, the effluent was carefully collected from the column at certain time intervals to assess the residual PFOA concentration with high accuracy. A schematic illustration of the column tests is depicted in Figure 1.

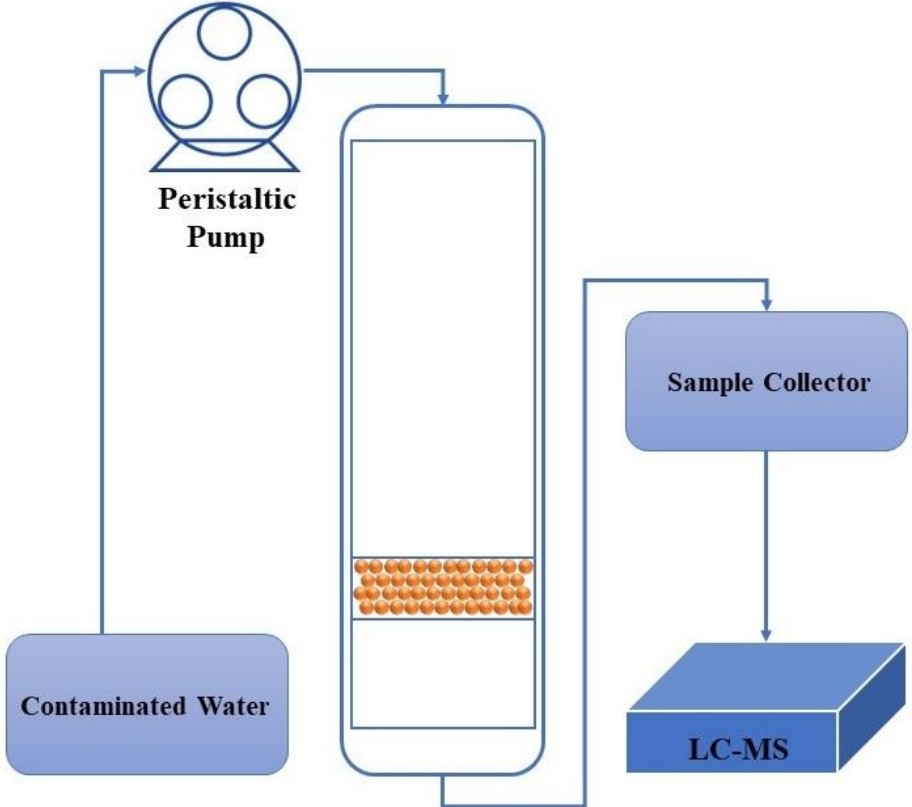

**Figure 1.** Schematic illustration of PFOA removal under continuous flow column text.

### 3. Result and Discussion

#### 3.1. Characterization

#### 3.1.1. PXRD

The current analysis aimed to explore the crystallographic properties of the synthesized MOF materials via PXRD analysis (Figure 2). The sharp and well-defined peaks observed at 7.14° (111) and 8.55° (002) exhibit high crystallinity of the Zr-MOF, indicating the presence of molecular layers possessing lamellar and intercalated frameworks [48]. Moreover, the intense peaks observed at 30.71°, 25.73°, 17.33°, 14.71°, and 11.87° are attributed to the (117), (006), (004), (222), and (022) planes, respectively [49]. The PXRD pattern obtained for La-MOF revealed the presence of sharp peaks at 23.97°, 21.48°, 19.68°, 18.02°, 15.19°, 14.26°, 8.80°, and 7.48°, and some peaks with lower intensities at 35.69°, 32.87°, and 28.95°, which are indicative of a well-ordered crystalline structure [50]. These results clearly indicate that the synthesized MOF materials possess well-defined crystalline structures, as confirmed by the characteristic PXRD peaks.

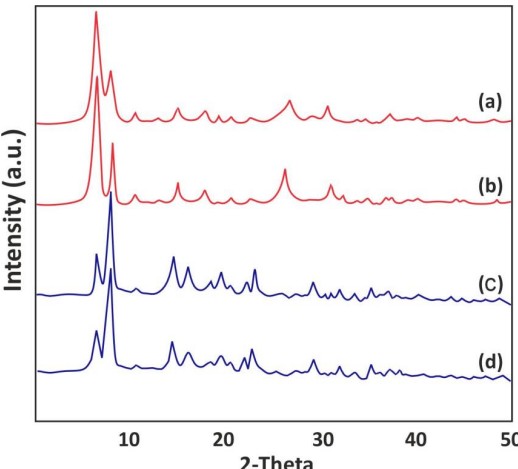

**Figure 2.** XRD pattern for (a) and (b) Zr-MOF after and before PFOA adsorption, and (c) and (d) La-MOF after and before PFOA adsorption.

### 3.1.2. FTIR

FTIR analysis was conducted to explore the functionalities present in the as-synthesized MOFs, with a range of wavenumbers spanning from 400 to 4000 cm$^{-1}$ (Figure 3). The FTIR spectrum of La-MOF displayed several characteristic peaks, including peaks at approximately 3430 cm$^{-1}$ attributed to the -OH stretching resulting from hydroxyl moieties that exist in the structure of MOF. Moreover, the vibrations ascribed to the aromatic carbons were confirmed with peaks at 1400, 1536, and 1588 cm$^{-1}$, and the distinctive peaks at 1411 and 1695 cm$^{-1}$ illustrate the symmetric and asymmetric vibrations of carboxylate moieties, respectively [51]. Furthermore, the peak at around 1395 cm$^{-1}$ shows the existence of the -OH group, and the characteristic peak at approximately 1013 cm$^{-1}$ validates the existence of -C-O-C stretching vibration. In addition, the peak at 751 cm$^{-1}$ confirmed the bending vibration of C-H in the benzene ring, and the peak at 512 cm$^{-1}$ suggested the modes of La-O bending vibrations [52]. Comparable peaks were observed in the FTIR spectra of Zr-MOF, indicating that practically identical functional groups were present in its structure. The peaks around 512 cm$^{-1}$ further suggested the modes of Zr-O bending vibrations [49].

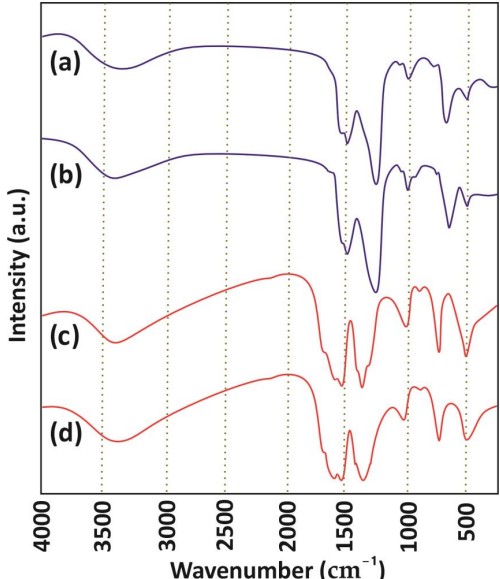

**Figure 3.** FTIR spectra for (a) and (b) La-MOF after and before PFOA adsorption, and (c) and (d) Zr-MOF after and before PFOA adsorption.

### 3.1.3. BET

The BET method was utilized to evaluate the surface area and porous structure of the as-synthesized MOF adsorbents; the corresponding data are recorded in Figure 4. Types I and IV hysteresis loops in both Zr- and La-MOF structures had comparable features. The outcomes verified that the produced MOFs contain a great number of mesopores on their surface. It is confirmed that a homogenous mesoporous network has formed in both MOFs, which are generally spread within 5 and 22 nm (Figure 4). The La-MOF and Zr-MOF materials have surface areas of 76.90 and 293.50 m$^2$/g, respectively. In contrast to previously reported works in the literature, a decrease in surface area was observed for the MOFs investigated in this study. For example, Abid et al. achieved a significantly higher surface area of 1433 m$^2$/g for Zr-MOF in their research than the Zr-MOF synthesized in this study [53]. Safinejad et al. synthesized an La-MOF using pure chemicals for drug delivery applications with a surface area of 521 m$^2$/g [54]. Usually, the surface areas obtained for Zr-MOF and La-MOF synthesized from pure chemicals are in the range of 790–2700 m$^2$/g and 240–530 m$^2$/g, respectively [55–58]. The development of a mesoporous framework and reduction in specific surface areas in both La- and Zr-MOFs are regarded as a consequence of the one-pot synthesis approach of converting PET into terephthalic acid to generate the MOFs. As a result, a significant proportion of MOFs are in the bulk phase.

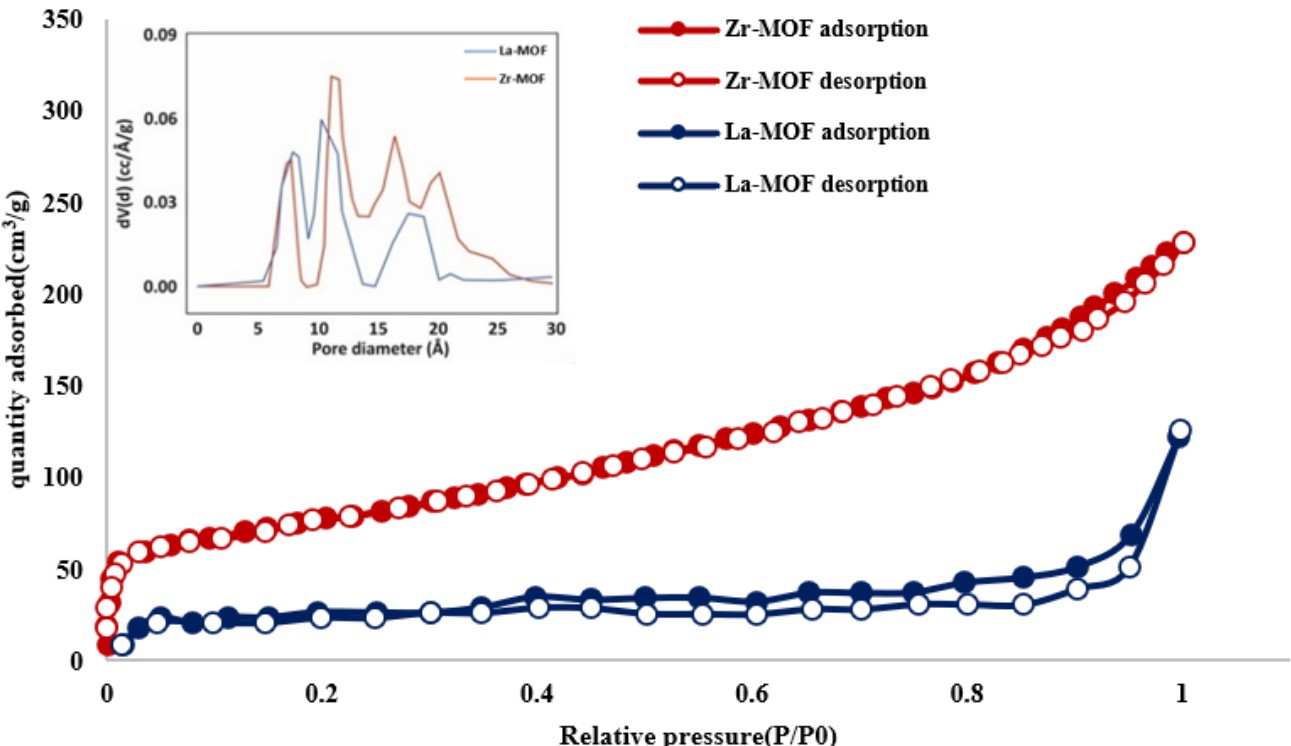

**Figure 4.** N$_2$ adsorption and desorption isotherms for La-MOF (blue points) and Zr-MOF (red points).

### 3.1.4. XPS

The photoelectron spectra characteristics for La-MOF are visible in the broad spectra at energies of 310, 556, and 862 eV, which correspond to C 1s, O 1s, and La 3d, respectively (Figure 5). The C 1s of La-MOF (Figure 5b) include some chemically displaced constituents, identified by a deconvolution as -COO-, C=O, and -C-C-, with associated peaks at 290.3, 286.4, and 284.6 eV. The C=O, free -OH, and La−O groups, which are present on the adsorbent surface, are associated with the peaks at 557.7, 555.2, and 554.1 eV, respectively (Figure 5d) [59]. As there are more carboxylate than hydroxyl groups present in this sample, the intensity of carboxylate oxygen is greater than that of hydroxyl oxygen. The detected peaks at 861.7 and 855.1 eV in the 3D core-level photoelectron spectroscopy

for La match the La 3d$_{3/2}$ and La 3d$_{5/2}$ orbitals. As demonstrated in Figure 6, the C 1s spectrum of the as-synthesized Zr-MOF unveiled deconvolution as C=O, C-O, O-C=O, and C-C at distinct binding energies of 288.2, 287.3, 289.5, and 285.4 eV, each representing various functionalities containing oxygen (Figure 6d) [60]. Four discernable peaks at 535.40, 533.60, 531.80, and 530.30 eV, respectively, corresponding to the Zr-O, O-C, O=C, and COO- moieties, are recorded in Figure 6c. The corresponding binding energies, atomic percentages, oxygen to carbon ratios, as well as metal to carbon ratios are given in Table 1. The metal-to-ligand ratio and synthesis conditions were identical for both MOFs. However, the composition analysis presented in Table 1 reveals that the La-MOF exhibits a higher percentage of metals compared to the Zr-MOF. It is reasonable to infer that a larger number of La metal ions in the La-MOF form coordination bonds with the ligands, leading to a higher concentration of La-ligand complexes within the La-MOF structure. Conversely, the lower percentage of Zr in the Zr-MOF indicates a relatively smaller number of Zr metal ions in contact with the ligands. This implies that certain ligands in the Zr-MOF may not be coordinated with Zr metal ions, resulting in the presence of uncoordinated, partially coordinated, or free ligands within its structure. The presence of these free ligands in the Zr-MOF could be attributed to factors such as incomplete coordination during synthesis or the presence of structural defects. Consequently, the repulsion between the uncoordinated ligands and PFOA can occur due to electrostatic forces or steric hindrance, which can account for the reduced adsorption capacity of the Zr-MOF compared to the La-MOF. Electrostatic repulsion arises from the repulsive forces experienced between the negatively charged carboxylic groups on both the ligands and PFOA. Steric hindrance occurs when the spatial arrangement of the ligands restricts the penetration of PFOA molecules into the MOF structure, limiting their entrapment. Furthermore, the XRD analysis confirms the aforementioned findings by indicating a lamellar structure for the Zr-MOF, while the La-MOF exhibits a well-ordered structure. This suggests that certain regions of the Zr-MOF may possess a bulk-like form, possibly due to the presence of mesopores or interlayer spaces between the layers. Despite its less ordered structure, the Zr-MOF retains a significant surface area attributed to these porous regions. The larger surface area of the Zr-MOF further supports the notion that repulsive forces among uncoordinated ligands and PFOA contribute to its reduced adsorption capacity compared to the La-MOF.

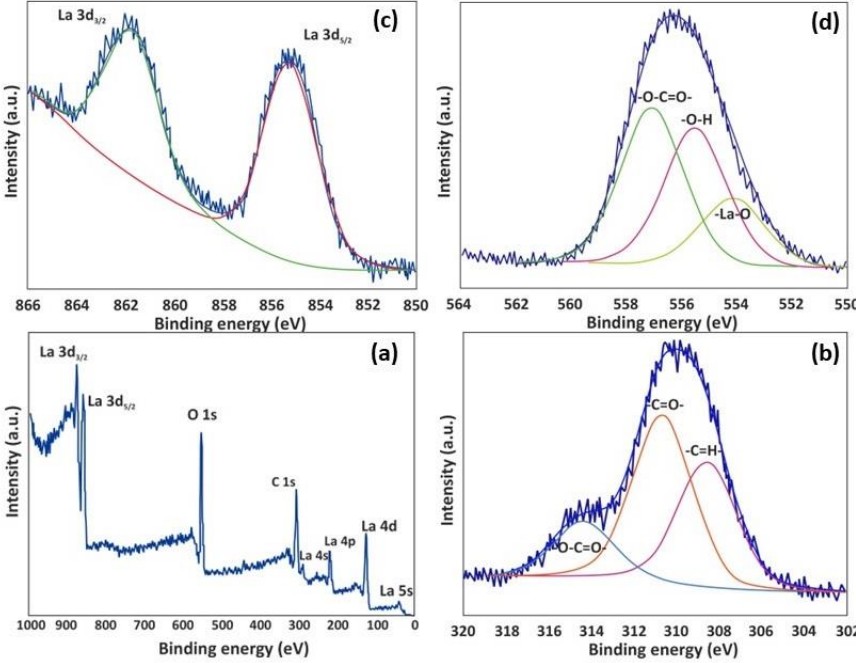

**Figure 5.** XPS spectra of La-MOF: (**a**) wide scan, (**b**) C 1s, (**c**) La 3d, (**d**) O 1s.

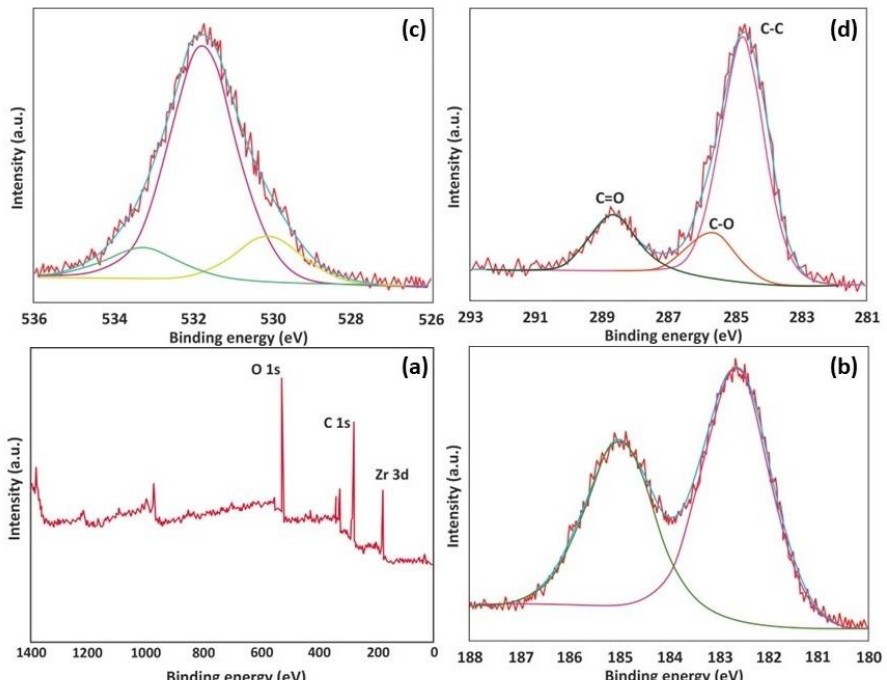

**Figure 6.** XPS spectra of Zr-MOF: (**a**) wide scan, (**b**) Zr 3d, (**c**) O 1s, (**d**) C 1s.

**Table 1.** Elemental analysis and binding energies for Zr-MOF and La-MOF.

| Adsorbent | Component | Binding Energy (eV) | Percentage (%) | Oxygen/Carbon Ratio | Metal/Carbon Ratio |
|-----------|-----------|---------------------|----------------|---------------------|--------------------|
|           | La        | 862                 | 14.51          |                     |                    |
| La-MOF    | O         | 556                 | 24.39          | 0.452               | 0.269              |
|           | C         | 310                 | 53.92          |                     |                    |
|           | Zr        | 182                 | 10.82          |                     |                    |
| Zr-MOF    | O         | 561                 | 30.12          | 0.515               | 0.185              |
|           | C         | 308                 | 58.42          |                     |                    |

### 3.1.5. TGA

The TGA curves of the adsorbents prepared in this study are depicted in Figure 7. As observed from the TGA curve of the La-MOF, the initial decline in weight at 330 °C can be attributed to carbonate conversion, while the second weight loss at 470 °C is linked to the formation of $La_2O_3$. The final weight reduction observed at temperatures above 645 °C can be attributed to the degradation of ligands associated with the metal centers. Conversely, the TGA curve of Zr-MOF demonstrated framework disintegration at 450 °C, leading to the production of $ZrO_2$. Notably, the TGA data reveals that La-MOF exhibits superior thermal stability in comparison to Zr-MOF.

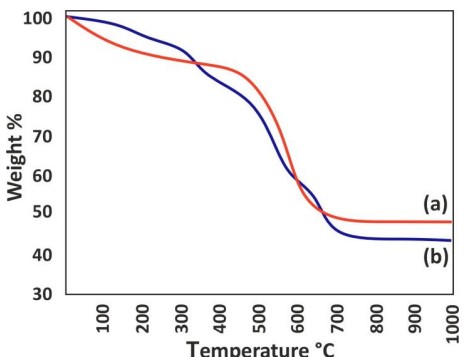

**Figure 7.** TGA curves: (a) Zr-MOF and (b) La-MOF.

### 3.2. Assessment of PFOA Adsorption, Isotherms, and Kinetics

The present study elucidated the adsorption potential of La- and Zr-MOFs synthesized from waste PET-based materials for PFOA removal using Freundlich and Langmuir isotherm equations as well as pseudo-1st and 2nd-order models for kinetic data investigation. According to Figure 8 and Table 2, the maximum theoretical adsorption capacity of PFOA for La-MOF and Zr-MOF was 364 and 340 mg/g, respectively, as determined by the Langmuir isotherm. However, the actual experimental maximum PFOA uptake was observed to be 290 mg/g for Zr-MOF and 310 mg/g for La-MOF, indicating that the real PFOA sorption performance of the materials was lower than the theoretical capacity. According to Figure 9 and Table 3, the adsorption behavior of PFOA followed the pseudo-2nd-order model, indicating that the rate-limiting step in the sorption process is chemisorption [2]. The adsorption isotherms on both Zr-MOF and La-MOF were in good agreement with the Langmuir model, demonstrating that the PFOA adsorption occurred at a homogeneous surface with a limited number of identical sorption nodes [2]. According to these results, the MOFs produced demonstrated promising potential as effective adsorbents for the removal of PFOA from water environments. Table 4 represents the performance of similar adsorbents for PFOA elimination from aqueous environments. For instance, Kong et al. [31] obtained 76.59 mg/g PFOA adsorption capacity, which is much lower than that of Zr- and La-MOF. However, there are some distinctive adsorbents that possess exceptional performance for PFOA removal. In the context of sustainability, our study goes beyond the performance comparison by highlighting the eco-friendly attributes of our synthesized MOFs. The fabricated MOFs demonstrate exceptional adsorption capacities for PFOA removal, allowing for efficient pollutant removal and reducing the need for large quantities of adsorbents. Our strategy promotes the recycling of waste materials and reduces the environmental impact.

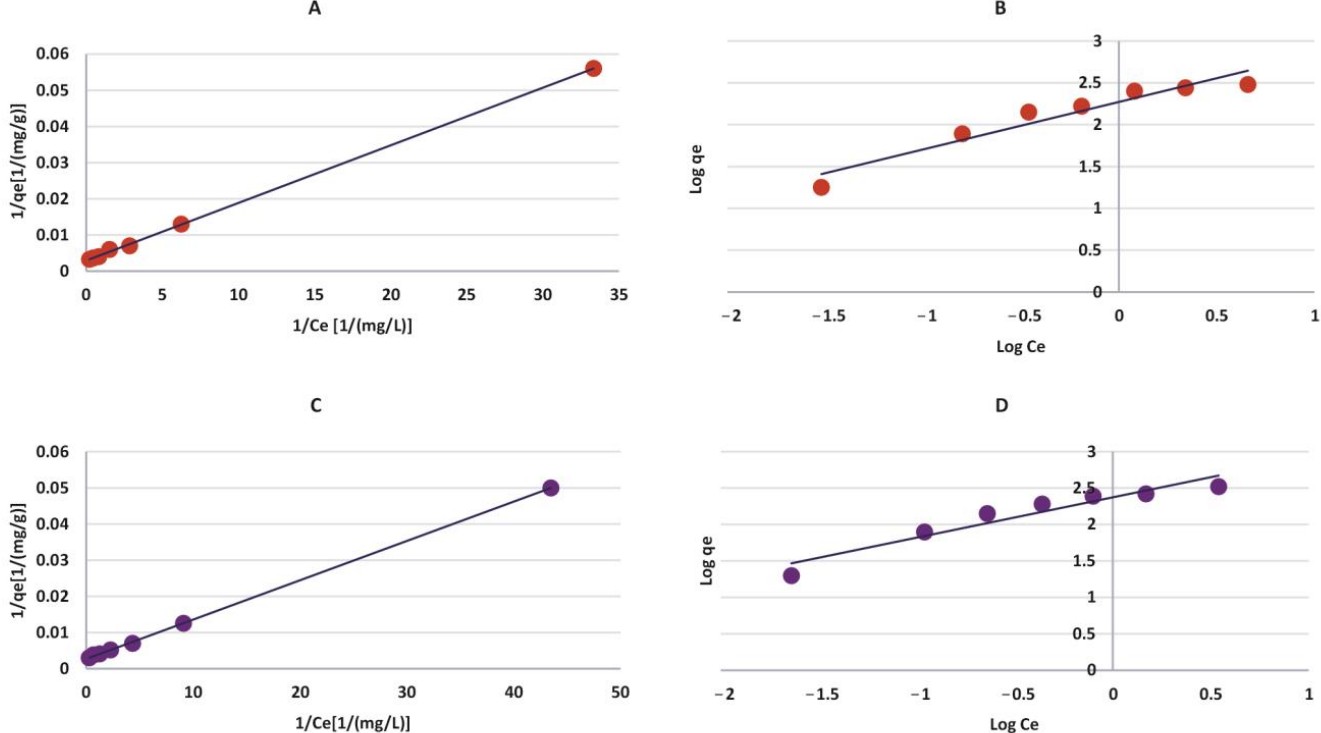

**Figure 8.** Adsorption isotherms of (**A**) Langmuir model and (**B**) Freundlich model for Zr-MOF, (**C**) Langmuir model and (**D**) Freundlich model for La-MOF.

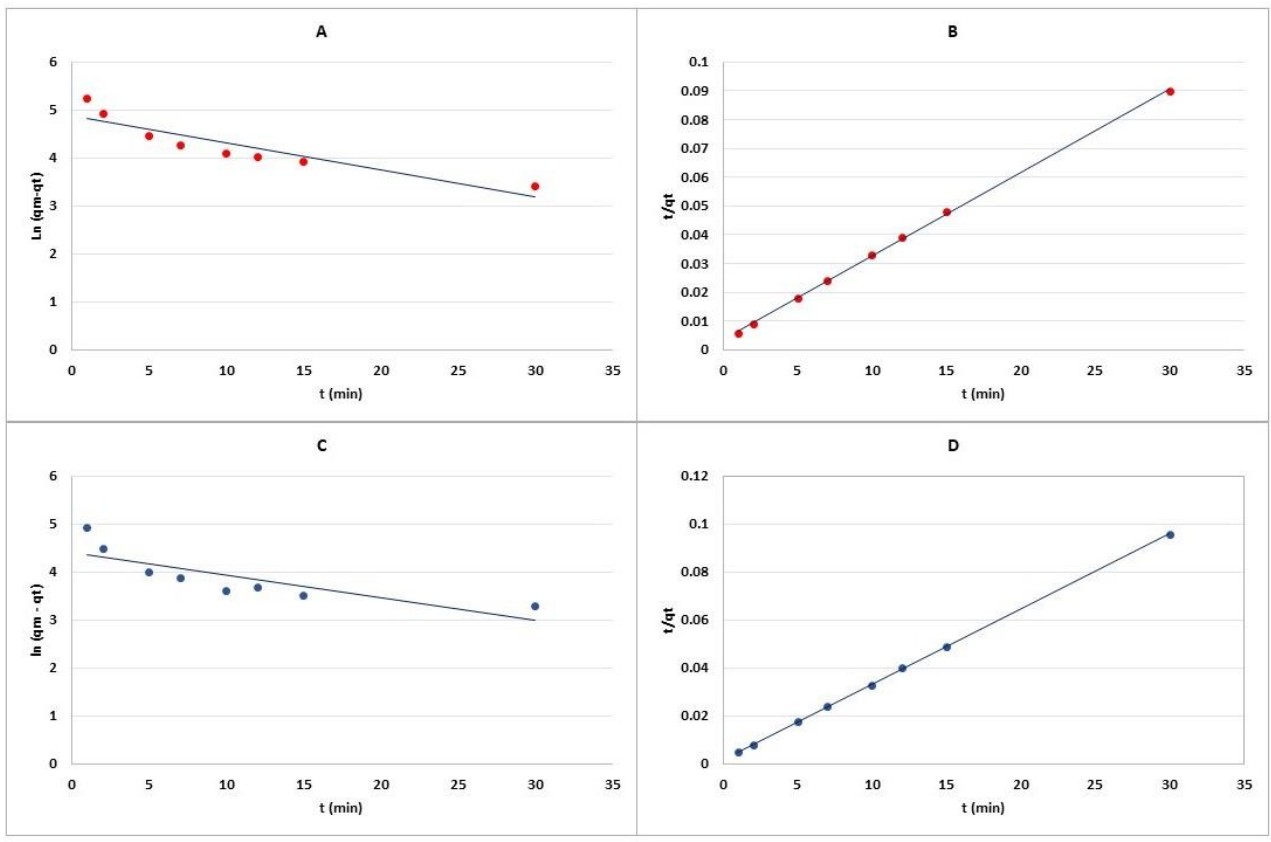

**Figure 9.** Adsorption kinetics of (**A**) pseudo-1st-order and (**B**) pseudo-2nd-order for Zr-MOF, and (**C**) pseudo-1st-order and (**D**) pseudo-2nd-order for La-MOF.

**Table 2.** Isotherm limitations of Zr-MOF and La-MOF.

|  | **Langmuir Model** | | | **Freundlich Model** | | |
|---|---|---|---|---|---|---|
|  | $q_m$ **(mg/g)** | $K_l$ **(L/mg)** | $R^2$ | $n$ | $K_f$ **(mg/g)** | $R^2$ |
| Zr-MOF | 340 | 1.8 | 0.999 | 1.76 | 186.9 | 0.922 |
| La-MOF | 364 | 2.5 | 0.999 | 1.81 | 236.15 | 0.91 |

**Table 3.** Kinetic limitations of La-MOF and Zr-MOF.

| **Materials** | **Pseudo-1st-Order** | | | **Pseudo-2nd-Order** | | |
|---|---|---|---|---|---|---|
|  | $q_m$ **(mg/g)** | $K_1$ **(1/min)** | $R^2$ | $q_m$ **(mg/g)** | $K_2$ **(g/mg·min)** | $R^2$ |
| Zr-MOF | 83.1 | 0.0475 | 0.6633 | 322.58 | 0.005 | 0.9999 |
| La-MOF | 131.63 | 0.0567 | 0.8243 | 332.7859 | 0.0032 | 0.9999 |

**Table 4.** Comparison of the PFOA adsorption capacity of some recently reported adsorbents.

| **Adsorbents** | **Adsorption Capacity (mg/g)** | **Reference** |
|---|---|---|
| F-TiO$_2$@MIL-125 | 76.59 | [31] |
| MIL-96-RHPAM2 | 340 | [61] |
| DUT-5-2 | 98.2 | [62] |
| MIL-101(Cr)-PAM | 492.7 | [37] |
| Al-MOF | 169.2 | [63] |
| La-MOF | 364 | Present work |
| Zr-MOF | 340 | Present work |

### 3.3. Assessment of the Initial pH of the Solution and PFOA Adsorption Mechanism

The initial pH value of a solution is a crucial parameter that significantly influences the adsorption mechanism of perfluorooctanoic acid onto La- and Zr-MOF. In the present study, the impact of the initial pH value on the adsorption of PFOA onto Zr-MOF and La-MOF was extensively investigated. According to Figure 10, the experimental results demonstrate that the best adsorption performance for both MOFs was observed at an initial pH of 3 and 3.7 for Zr-MOF and La-MOF, respectively. This observation can be attributed to the point of zero charge (PZC) of MOFs. The PZC of Zr-MOF was found to be at pH 4.8, while for La-MOF, it was 5.7, indicating that at a pH lower than their PZC, the MOFs' surfaces become positively charged and thus favor the adsorption of PFOA [64].

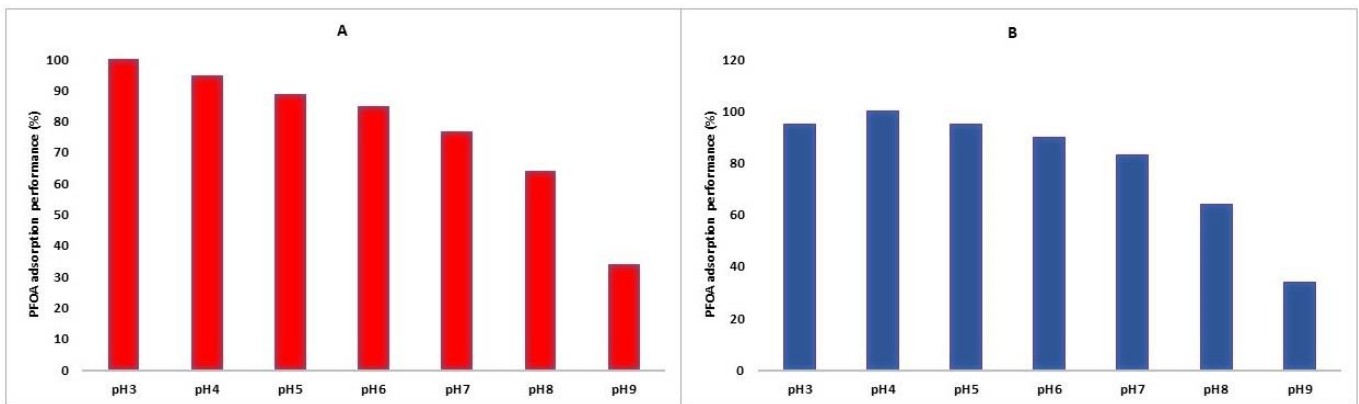

**Figure 10.** PFOA removal at a pH range of 3–9 for (**A**) Zr-MOF (**B**) La-MOF.

Regarding the adsorption mechanism, electrostatic interactions and/or hydrophobic interactions are believed to be the main driving forces for PFOA adsorption onto MOFs [65]. The absence of significant changes in the intensity of peaks in the X-ray diffraction spectra of both MOFs after adsorption, along with the observation of no changes in the intensity of peaks in the Fourier-transform infrared spectra of the MOFs, strongly suggests that Lewis acid-base complexation between the $Zr^{4+}$ and $La^{3+}$ clusters and the headgroup of PFOA is unlikely to occur. These observations are consistent with previous works that have reported that PFOA adsorption onto MOFs is primarily driven by electrostatic and/or hydrophobic interactions. Overall, the results suggest that the initial pH value of the solution plays a significant role in PFOA adsorption onto MOFs, and electrostatic and/or hydrophobic interactions are the primary mechanisms driving PFOA adsorption onto MOFs.

### 3.4. Utilization of MOF Materials in Practical Settings

This investigation sought to assess the efficacy of newly synthesized MOF materials in PFOA elimination from water under continuous flow conditions (600 µg/L) via fixed-bed column experiments. The results of this study demonstrate that La-MOF demonstrated significantly superior adsorption capability when compared to Zr-MOF examined in this research, according to Figure 11. La- and Zr-MOF were found to have breakthrough positions of 174 and 150 min, respectively, based on the PFOA limit of 0.07 µg/L in drinking water [66]. In terms of bed volume, these figures correspond to 522 mL and 452 mL, respectively. In comparison to recently developed adsorbents such as leaf biomass and $Fe_3O_4$-biochar hybrids, which showed breakthrough times of less than 100 min, the MOFs synthesized in this study exhibited remarkable performance, highlighting their viability for practical applications [67,68].

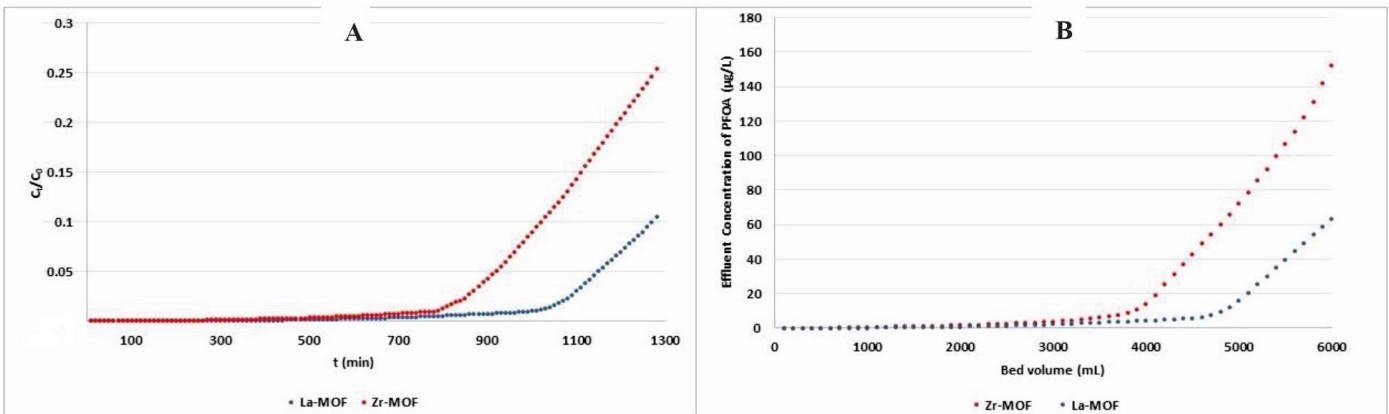

**Figure 11.** Breakthrough of PFOA adsorption on Zr-MOF and La-MOF based on (**A**) time and (**B**) Bed volume.

Adsorbent regeneration is critical for developing an affordable adsorption process. According to Figure 12, the PFOA removal capability of the produced MOFs slightly decreased with an increasing number of recycling sessions. This might be ascribed to pore volume loss throughout the subsequent adsorption-desorption stage, as some PFOA particles may become trapped in the adsorbent structure and resist removal even after regeneration. Nevertheless, the PFOA removal performance of the as-synthesized MOFs for PFOA remained relatively stable, decreasing by less than 10% for up to seven cycles. The integration of sustainability principles in our study extends beyond the use of recycled materials. The stability of the MOFs over multiple cycles of PFOA adsorption and desorption showcases their durability and potential for long-term use, minimizing waste generation and ensuring prolonged effectiveness.

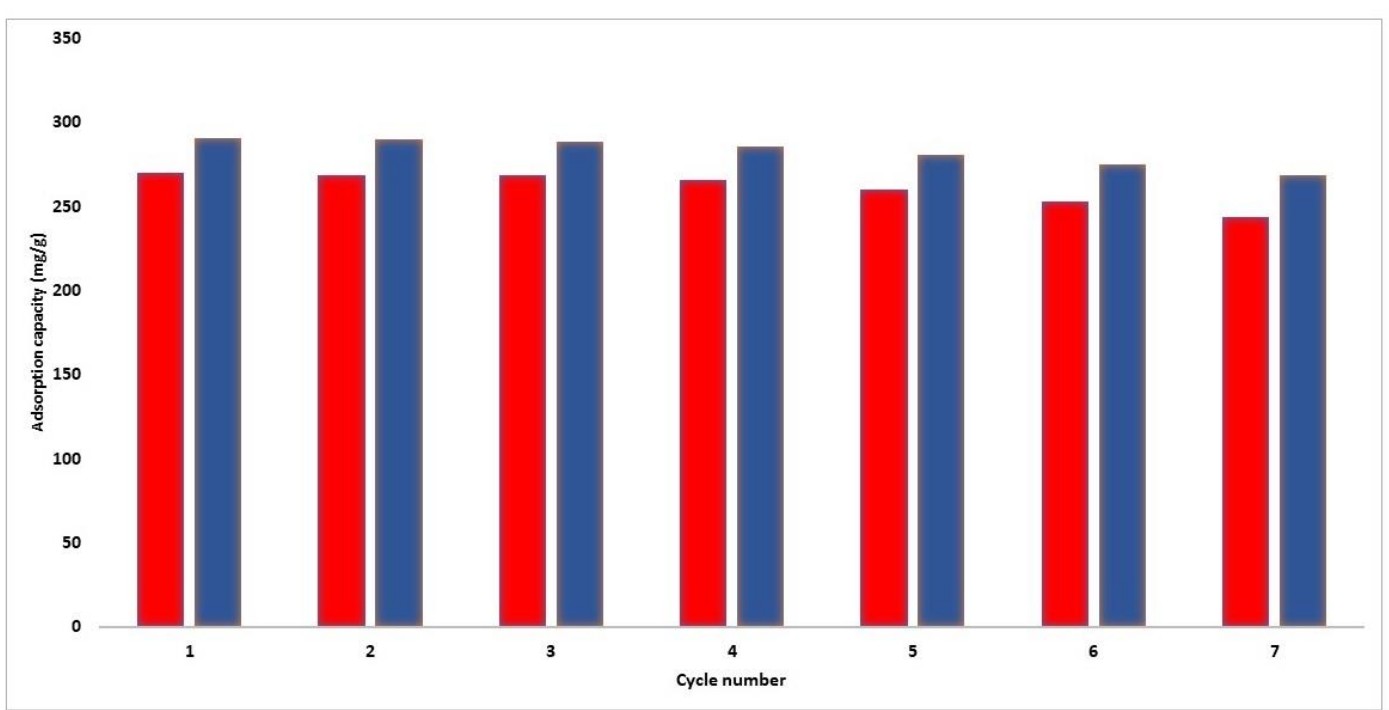

**Figure 12.** Adsorption capacity at different cycles: Zr−MOF (red), La−MOF (blue).

## 4. Conclusions

In conclusion, our study has demonstrated the feasibility of transforming waste materials into valuable MOFs for the sustainable removal of PFOA, an organic pollutant persistent to biodegradation and traditional water treatment methods, from aqueous solutions. Through the utilization of readily available PET bottles as a source of terephthalic acid (BDC) for MOF synthesis, we were able to reduce production costs and promote an eco-friendly solution to PET waste. Our one-pot microwave-assisted synthesis strategy for MOF fabrication facilitated a simplified, more efficient process that resulted in high-quality MOF materials, La-MOF and Zr-MOF. The resulting La-MOF and Zr-MOF materials exhibit impressive surface areas of 76.90 and 293.50 $m^2/g$, respectively, with enhanced thermal stability observed in La-MOF up to 470 °C and Zr-MOF up to 450 °C. The maximum experimental PFOA adsorption capacities of 310 mg/g for La-MOF and 290 mg/g for Zr-MOF highlight their effectiveness in removing PFOA from contaminated water. Our findings further suggest that both La- and Zr-MOF follow the pseudo-2nd-order kinetic model and the Langmuir isotherm for PFOA removal while maintaining stable adsorption efficiency for up to seven cycles. The real-world application of these materials was also investigated, revealing that in fixed-bed column tests, breakthrough positions of 150 and 174 min were observed for Zr- and La-MOF, respectively, with corresponding bed volumes of 452 mL and 522 mL based on the PFOA limit of 0.07 μg/L. Eventually, our approach not only circumvented conventional PET recycling practices but also facilitated the production of high-quality MOFs that demonstrated excellent performance in removing PFOA from water. The synthesis of MOFs from waste PET bottles via a one-pot microwave-assisted approach has the potential to pave the way for the sustainable and efficient treatment of organic pollutants in water environments.

**Author Contributions:** Conceptualization, A.E.J.; Methodology, A.E.J., R.D.C.P., M.A., A.E. and C.W.; Software, M.A. and A.E.; Validation, A.E.J., M.A., A.E., C.W. and S.S.S.; Formal analysis, A.E.J., C.W. and S.S.S.; Investigation, A.E.J., R.D.C.P., M.A. and C.W.; Data curation, S.S.S. and H.H.T.; Writing—original draft, A.E.J.; Writing—review & editing, R.D.C.P., M.T.C.A., M.A., A.E., S.S.S. and H.H.T.; Visualization, A.E. and S.S.S.; Supervision, A.E.J., R.D.C.P., M.T.C.A. and C.W. All authors have read and agreed to the published version of the manuscript.

**Funding:** This research was funded by the Deanship of Scientific Research at King Khalid University under grant number RGP.2/57/44.

**Informed Consent Statement:** Not applicable.

**Data Availability Statement:** The data that support the findings of this study are available on request from the corresponding author.

**Acknowledgments:** This work was supported by the King Khalid University, Abha, Saudi Arabia. The authors extend their appreciation to the Deanship of Scientific Research at King Khalid University for funding this work through the Large Groups Project under grant number (R.G.P. 2/57/44).

**Conflicts of Interest:** The authors declare no conflict of interest.

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
