# Peer review of "Transforming Waste into Value: Eco-Friendly Synthesis of MOFs for Sustainable PFOA Remediation"

_sustainability, doi:10.3390/su151310617_

Round 1

Reviewer 1 Report

This manuscript introduced an approach to synthesize La-MOF and Zr-MOF from PET bottle for PFOA removal, which is a sustainable solution to recycle waste plastic into useful material and application. The authors have shown their ability to fabricate high BET area MOFs (in comparison to literature) and some prospective results in PFOA removal.

However, this manuscript will require a significant revision in structure, formatting and proof-reading. For instance,
+ Only the names of characterization methods were mentioned without any detail of analytical conditions. LCMS method and the regeneration of used MOFs were not mentioned.
+ Too many general statements were repeated throughout the manuscript. The conclusion was also too general without much key numerical results.
+ Most of the results were not cross-referenced to relevant tables and figures.
+ The acronyms were not strictly followed the formal writing style (e.g., the acronyms of characterisation methods were introduced twice, the full name of "metal organic framworks" were used multiple times although MOFs had been introduced, etc.)
The detail comments can be found in the attachment.

The manuscript was written in high proficiency English. Some editing is suggested for acronyms, repeating sentences/phrases, etc. (please see the attachment).

Author Response

Response to reviewer #1

Reviewer 1 comments and suggestions for authors:

This manuscript introduced an approach to synthesize La-MOF and Zr-MOF from PET bottle for PFOA removal, which is a sustainable solution to recycle waste plastic into useful material and application. The authors have shown their ability to fabricate high BET area MOFs (in comparison to literature) and some prospective results in PFOA removal.

However, this manuscript will require a significant revision in structure, formatting and proof-reading. For instance,

Comment: Only the names of characterization methods were mentioned without any detail of analytical conditions. LCMS method and the regeneration of used MOFs were not mentioned.

Authors' response to comment: Thank you for your comment. Section 2.3. was completely revised with explanation regarding characterization methods. Also, section 2.5 was added regarding the LC-MS analysis and section 2.8 was added to explain the regeneration of used MOFs.

Please see lines 152 to 171.

Please see lines 191 to 203.

Please see lines 236 to 245.

The added text is as below:

“2.3. Characterization Techniques

TGA was performed utilizing the PerkinElmer Pyris Diamond, a thermogravimetric analyzer. The temperature ramp ranged from ambient conditions to 800°C, with a heating rate of 10°C/min, all while maintaining a nitrogen gas flow. Throughout the experiment, the weight loss and thermal decomposition characteristics of Zr-MOF and La-MOF were tracked across varying temperatures. For nitrogen adsorption-desorption isotherm measurements, an automated gas adsorption system, the Micromeritics TriStar II Plus, was employed. The measurements were conducted at a temperature of -196°C. Prior to the analysis, the samples underwent degassing for a duration of 4 hours under vacuum conditions at 200°C. As part of the analysis process, FTIR analysis was conducted at different stages on the Zr-MOF and La-MOF. Spectra in the range of 4000-400 cm-1 were recorded utilizing the PerkinElmer Spectrum Two spectrometer, which possessed a resolution of 4 cm-1. To prepare the FTIR samples, the materials were mixed with KBr and compressed into pellets. To determine the elemental composition and chemical state of the materials, XPS spectra were recorded using the Thermo Fisher Scientific K-Alpha+ spectrometer. The samples were excited with monochromatic Al Kα X-ray source (hν = 1486.6 eV), and the emitted photoelectrons were meticulously analyzed to gather information regarding the surface chemistry of the materials. Lastly, XRD patterns were obtained using the Bruker D8 Advance diffractometer equipped with Cu Kα radiation. The instrument operated at 30 mA and 40 kV, scanning the 2θ range from 2° to 50° with a step size of 0.02°.

2.5. Liquid Chromatography Mass Spectroscopy (LC-MS) Analysis:

LC-MS analysis was conducted to enable comprehensive quantification of PFOA. The analysis was carried out using an LC-MS system, comprising an Agilent 1260 Infinity II series high-performance liquid chromatography (HPLC) instrument coupled with an Agilent 6470 Triple Quadrupole Mass Spectrometer. This setup facilitated precise identification and quantification of PFOA. For chromatographic separation, an Agilent Zorbax Eclipse XDB-C18 column was utilized. The column provided efficient separation and resolution of PFOA. The LC mobile phase consisted of a solvent mixture of high-purity methanol and water, prepared using HPLC-grade solvents. This solvent system ensured optimal chromatographic performance and reproducibility. During the LC-MS analysis, PFOA was detected and quantified based on its specific mass-to-charge ratio (m/z) and retention time. Calibration curves were constructed using known concentrations of PFOA standards to accurately determine the concentration of PFOA in the samples.

2.8. Desorption procedure of PFOA from MOF surface:

To regenerate the PFOA-loaded Zr-MOF and La-MOF adsorbents and facilitate the desorption of PFOA from their surfaces, a regeneration procedure using NaOH was employed. The PFOA-loaded MOF samples were immersed in a 0.5 M NaOH solution, ensuring complete coverage of the MOF surfaces. Gentle stirring was applied for 24 hours, under controlled room temperature conditions. After the desorption process, the MOF samples were separated from the NaOH solution via centrifugation. Thorough rinsing with deionized water was performed to remove any remaining NaOH and PFOA residues. The regenerated MOF samples were then oven-dried at 100°C to remove excess water.

Comment: Too many general statements were repeated throughout the manuscript. The conclusion was also too general without much key numerical results.

Authors' response to comment: Thank you for your comment. It was corrected. The conclusion section was revised according to your suggestion. Quantitative results were added to the conclusion.

Please see this section.

 The revised conclusion is as follow:

“In conclusion, our study has demonstrated the feasibility of transforming waste materials into valuable MOFs for the sustainable removal of PFOA, an organic pollutant persistent to biodegradation and traditional water treatment methods, from aqueous solutions. Through the utilization of readily available PET bottles as a source of terephthalic acid (BDC) for MOF synthesis, we were able to reduce production costs and promote an eco-friendly solution to PET waste. Our one-pot microwave-assisted synthesis strategy for MOF fabrication facilitated a simplified, more efficient process that resulted in high-quality MOF materials, La-MOF and Zr-MOF. The resulting La-MOF and Zr-MOF materials exhibit impressive surface areas of 76.90 and 293.50 m2/g, respectively, with enhanced thermal stability observed in La-MOF up to 470 °C and Zr-MOF up to 450 °C. The maximum experimental PFOA adsorption capacities of 310 mg/g for La-MOF and 290 mg/g for Zr-MOF highlight their effectiveness in removing PFOA from contaminated water. Our findings further suggest that both La- and Zr-MOF follow pseudo-2nd-order kinetic model and the Langmuir isotherm for PFOA removal, while maintaining stable adsorption efficiency for up to seven cycles. The real-world application of these materials was also investigated revealing that in fixed-bed column tests, breakthrough positions of 150 and 174 minutes were observed for Zr- and La-MOF, respectively, with corresponding bed volumes of 452 mL and 522 mL based on the PFOA limit of 0.07 µg/L. Eventually, our approach not only circumvented the conventional PET recycling practices, but also facilitated the production of high-quality MOFs that demonstrated excellent performance in removing PFOA from water. The synthesis of MOFs from waste PET bottles via a one-pot microwave-assisted approach has the potential to pave the way for the sustainable and efficient treatment of organic pollutants in water environments.”

Comment: Most of the results were not cross-referenced to relevant tables and figures.

Authors' response to comment: Thank you for your comment. The cross-references were added to the sentence as you suggested.

Please see lines 377, 382, 412, and 439.

Comment: The acronyms were not strictly followed the formal writing style (e.g., the acronyms of characterisation methods were introduced twice, the full name of "metal organic framworks" were used multiple times although MOFs had been introduced, etc.)

Authors' response to comment: Thank you for your comment. It was corrected.

The detail comments can be found in the attachment.

Comment 1: systems?.

Authors' response to comments 1: Thank you for your comment. It was corrected.

Please see line 47.

Comment 2: Recently sounds more formal.

Authors' response to comments 2: Thank you for your comment. It was corrected.

Please see line 51.

Comment 3: This sentence is kind of repeating

Authors' response to comments 3: Thank you for your comment. The sentence was deleted.

Comment 4: Suggest to use “with negligible effect of the C-F bonds”

Authors' response to comments 4: Thank you for your comment. Your suggestion was applied, and the sentence was corrected.

Please see lines 60 to 61.

Comment 5: Such as? Are they more dangerous than PFOA.

Authors' response to comments 5: Thank you for your comment. An explanation regarding the hazardous by-products of PFOA degradation via oxidative methods was added to the text.

Please see lines 63 to 73.

The added text is as below:

“For example, when long-chain PFAS compounds like PFOA are degraded using oxidation methods, they can be transformed into shorter-chain PFAS molecules such as perfluoropentanoic acid (PFPeA) and perfluorobutanoic acid (PFBA). These shorter-chain PFAS molecules exhibit higher solubility and stability in water compared to their long-chain counterparts. Consequently, the treatment of long-chain PFAS compounds using oxidative methods poses an additional challenge of effectively removing these newly formed short-chain PFAS molecules from the water or wastewater streams. Additionally, alternative methods such as sonolysis and photolysis, which employ sound waves or light for the degradation of PFAS compounds, along with the utilization of fungal enzymes, are currently in the experimental or developmental stage. The practical feasibility of implementing these approaches on a large scale has not yet been fully demonstrated.”

Comment 6: This paragraph summarizes some technologies

Authors' response to comments 6: Thank you for your comment. Your suggestion was applied to the manuscript.  

Please see lines 63 to 73, and 75 to 80.

The added text is as below:

“For example, when long-chain PFAS compounds like PFOA are degraded using oxidation methods, they can be transformed into shorter-chain PFAS molecules such as perfluoropentanoic acid (PFPeA) and perfluorobutanoic acid (PFBA). These shorter-chain PFAS molecules exhibit higher solubility and stability in water compared to their long-chain counterparts. Consequently, the treatment of long-chain PFAS compounds using oxidative methods poses an additional challenge of effectively removing these newly formed short-chain PFAS molecules from the water or wastewater streams. Additionally, alternative methods such as sonolysis and photolysis, which employ sound waves or light for the degradation of PFAS compounds, along with the utilization of fungal enzymes, are currently in the experimental or developmental stage. The practical feasibility of implementing these approaches on a large scale has not yet been fully demonstrated.

The filtration technologies employed for PFAS removal largely stem from existing filtration methods that were initially developed for addressing other contaminants. This implies that the feasibility of these approaches for large-scale applications has been established. However, the unique chemical characteristics of PFAS compounds and their behavior in water necessitate careful consideration when devising new separation techniques specifically tailored for PFAS treatment.”

Comment 7: What adsorbent materials have been reported in the literature.

Authors' response to comments 7: Thank you for your comment. Some examples of adsorbent materials are added to the manuscript.

Please see lines 84 to 85.

The added text is as below:

“such as granular activated carbon, clay, functionalized synthetic fibers, and silica zeolites”

Comment 8: Suggest to remove citation here.

Authors' response to comments 8: Thank you for your comment. The citation was removed as you suggested.

Comment 9: Better to clarify more about the PET sources.

Authors' response to comments 9: Thank you for your comment. The requested information regarding PET bottles was added to the manuscript.

Please see lines 135 to 137.

The added text is as below:

“Transparent PET bottles were gathered from local sources, and their lids were removed prior to washing. Subsequently, the bottles were manually cut into pieces approximately 2mm in size using scissors.”

Comment 10: The chemical formula should be introduced.

Authors' response to comments 10: Thank you for your comment. The chemical formulas were added to each chemical in section 2.1.

Please see lines 132 to 133.

Comment 11: This section it too general without detailed experimental condition

Authors' response to comments 11: Thank you for your comment. Section 2.3. was completely revised with explanation regarding characterization methods.

Please see lines 152 to 171.

The added text is as below:

“2.3. Characterization Techniques

TGA was performed utilizing the PerkinElmer Pyris Diamond, a thermogravimetric analyzer. The temperature ramp ranged from ambient conditions to 800°C, with a heating rate of 10°C/min, all while maintaining a nitrogen gas flow. Throughout the experiment, the weight loss and thermal decomposition characteristics of Zr-MOF and La-MOF were tracked across varying temperatures. For nitrogen adsorption-desorption isotherm measurements, an automated gas adsorption system, the Micromeritics TriStar II Plus, was employed. The measurements were conducted at a temperature of -196°C. Prior to the analysis, the samples underwent degassing for a duration of 4 hours under vacuum conditions at 200°C. As part of the analysis process, FTIR analysis was conducted at different stages on the Zr-MOF and La-MOF. Spectra in the range of 4000-400 cm-1 were recorded utilizing the PerkinElmer Spectrum Two spectrometer, which possessed a resolution of 4 cm-1. To prepare the FTIR samples, the materials were mixed with KBr and compressed into pellets. To determine the elemental composition and chemical state of the materials, XPS spectra were recorded using the Thermo Fisher Scientific K-Alpha+ spectrometer. The samples were excited with monochromatic Al Kα X-ray source (hν = 1486.6 eV), and the emitted photoelectrons were meticulously analyzed to gather information regarding the surface chemistry of the materials. Lastly, XRD patterns were obtained using the Bruker D8 Advance diffractometer equipped with Cu Kα radiation. The instrument operated at 30 mA and 40 kV, scanning the 2θ range from 2° to 50° with a step size of 0.02°.”

Comment 12: This section wasn’t at good structure

Authors' response to comments 12: Thank you for your comment. Section 2.4. was divided into three sections, including 2.4, 2.6, and 2.7.

Comment 13: How the PFOA solution was made?

Authors' response to comments 13: Thank you for your comment. The preparation of PFOA solution is added to the manuscript in section 2.4.

Please see lines 174 to 176.

The added text is as below:

“To create the stock solution of PFOA, 105.26 mg of PFOA powder (95%) was dissolved in 1 liter of water and thoroughly mixed until complete dissolution was achieved.”

Comment 14: Please add a section for LC-MS analysis

Authors' response to comments 14: Thank you for your comment. Section 2.5 was added regarding the LC-MS analysis.

Please see lines 191 to 203.

The added text is as below:

“2.5. Liquid Chromatography Mass Spectroscopy (LC-MS) Analysis:

LC-MS analysis was conducted to enable comprehensive quantification of PFOA. The analysis was carried out using an LC-MS system, comprising an Agilent 1260 Infinity II series high-performance liquid chromatography (HPLC) instrument coupled with an Agilent 6470 Triple Quadrupole Mass Spectrometer. This setup facilitated precise identification and quantification of PFOA. For chromatographic separation, an Agilent Zorbax Eclipse XDB-C18 column was utilized. The column provided efficient separation and resolution of PFOA. The LC mobile phase consisted of a solvent mixture of high-purity methanol and water, prepared using HPLC-grade solvents. This solvent system ensured optimal chromatographic performance and reproducibility. During the LC-MS analysis, PFOA was detected and quantified based on its specific mass-to-charge ratio (m/z) and retention time. Calibration curves were constructed using known concentrations of PFOA standards to accurately determine the concentration of PFOA in the samples.”

Comment 15: Suggest the author to use “Math” function

Authors' response to comments 15: Thank you for your comment. Equations were corrected according to your suggestion.

Comment 16: What n stands for?

Authors' response to comments 16: Thank you for your comment. A sentence defining the parameters in Langmuir and Freundlich isotherms was added to the manuscript in section 2.6.

Comment 17: How and how much of the sample was collected at each time point?

Authors' response to comments 17: Thank you for your comment. The volume of the samples did not change because the experiment was conducted on different samples. Typically, a series of samples were made, and the adsorption process was performed on each sample separately for varying time intervals. This explanation was also mentioned in the manuscript.

Comment 18: What was the volume of PFOA solution?

Authors' response to comments 18: Thank you for your comment. The volume of the sample was 100 mL. It is worth mentioning that the experimental conditions regarding pH study were added in section 2.7.

Please see line 231.

Comment 19: Can authors provide a justification why

Authors' response to comments 19: Thank you for your comment. To assess the maximum adsorption capacity of the adsorbents, a concentration range of 1-20 mg/L was utilized. However, it is important to acknowledge that PFOA concentrations in real contaminated water are typically lower. Therefore, a concentration of 600 μg/L, which closely resembles real-world scenarios, was chosen for practical settings.

Regarding the pH studies, the adsorption process was investigated across the entire concentration range. However, due to the extensive data obtained, only the results pertaining to a PFOA concentration of 10 mg/L, selected as a midpoint, were presented in the current paper. This concentration serves as a representative data point to illustrate the influence of pH on the adsorption process.

Comment 20: Should (a) and (b) be Zr-MOF instead?

Authors' response to comments 20: Thank you for your comment. The caption of figure was corrected. 

Please see lines 274 to 275.

Comment 21: How the surface areas compared with other similar studies in literature?

Authors' response to comments 21: Thank you for your comment. A text regarding the comparison of surface areas with previous similar works was added to the manuscript. 

Please see lines 302 to 305.

The added text is as below:

“In contrast to previously reported works in the literature, a decrease in surface area was observed for the MOFs investigated in this study. For example, Abid et al. achieved a significantly higher surface area of 1433 m2/g for Zr-MOF in their research.”

Comment 22: Suggest to include the heating rate for TGA study?

Authors' response to comments 22: Thank you for your comment. It is mentioned in section 2.3.

Please see line 155.

Comment 23: If the data were obtained from Fig 7

Authors' response to comments 23: Thank you for your comment. The cross-reference was added to the manuscript and table numbering was changed as you suggested.

Please see lines 359 and 377.

Comment 24: Similar to the previous comment

Authors' response to comments 24: Thank you for your comment. The cross-reference was added to the sentence as you suggested.

Please see line 382.

Comment 25: What specific MOF was presented in this study?

Authors' response to comments 25: Thank you for your comment. An aluminum-based MOF was investigated in this study. It was corrected on the table.

Please see table 4.

Comment 26: The figure should be meaningful

Authors' response to comments 26: Thank you for your comment. As the scale of these figures are logarithmic and fractional, it is not appropriate to write the name of Ce and qe. However, the units regarding each axis were added to axis titles for clarification.

Comment 27: As above, please cross-reference to the related Table/Figure

Authors' response to comments 27: Thank you for your comment. The cross-reference was added to the text as you suggested.

Please see line 412.

Comment 28: Not sure how authors got this number from Fig 9.

Authors' response to comments 28: Thank you for your comment. At first, the best performance of the La-MOF was observed at pH 3 and 4. To find the optimal pH value, the adsorption process was performed at pH values between 3 and 4. Therefore, it was found that the best pH value for La-MOF was 3.7. However, to avoid confusion about the data between pH 3-4 and better present the results column chart was used.

Comment 29: Not sure how the authors can conclude

Authors' response to comments 29: Thank you for your comment. The pH values of 4.8 and 5.7 were the point of zero charge values obtained from the PZC plots. The method of finding pH 3.7 was stated in comment 25.

Comment 30: Please add the cross-reference

Authors' response to comments 30: Thank you for your comment. The cross-reference was added to the text as you suggested.

Please see line 439.

Comment 31: How these numbers compared to literature?

Authors' response to comments 31: Thank you for your comment. As your suggestion, a comparison with recently developed adsorbents regarding the fixed-bed column tests was added to the manuscript.

Please see lines 442 to 445.

The added text is as below:

“In comparison to recently developed adsorbents like leaf biomass and Fe3O4-biochar hybrid, which showed breakthrough times of less than 100 minutes, the MOFs synthesized in this study exhibited remarkable performance, highlighting their viability for practical applications.”

Comment 32: How the used MOFs were recycled?

Authors' response to comments 32: Thank you for your comment. Section 2.8 was added to explain the recycling procedure.

Please see lines 236 to 245.

The added text is as below:

“2.8. Desorption procedure of PFOA from MOF surface:

To regenerate the PFOA-loaded Zr-MOF and La-MOF adsorbents and facilitate the desorption of PFOA from their surfaces, a regeneration procedure using NaOH was employed. The PFOA-loaded MOF samples were immersed in a 0.5 M NaOH solution, ensuring complete coverage of the MOF surfaces. Gentle stirring was applied for 24 hours, under controlled room temperature conditions. After the desorption process, the MOF samples were separated from the NaOH solution via centrifugation. Thorough rinsing with deionized water was performed to remove any remaining NaOH and PFOA residues. The regenerated MOF samples were then oven-dried at 100°C to remove excess water.

Comment 33: Using MOFs instead.

Authors' response to comments 33: Thank you for your comment. The “MOFs” word was used instead as you suggested.

Please see line 466.

Comment 34: PFOA instead.

Authors' response to comments 34: Thank you for your comment. The “PFOA” word was used instead as you suggested.

Please see line 466.

Comment 35: The conclusion is too general…

Authors' response to comments 35: Thank you for your comment. The conclusion section was revised according to your suggestion. Quantitative results were added to the conclusion.

Please see this section. The revised conclusion is as follow:

“In conclusion, our study has demonstrated the feasibility of transforming waste materials into valuable MOFs for the sustainable removal of PFOA, an organic pollutant persistent to biodegradation and traditional water treatment methods, from aqueous solutions. Through the utilization of readily available PET bottles as a source of terephthalic acid (BDC) for MOF synthesis, we were able to reduce production costs and promote an eco-friendly solution to PET waste. Our one-pot microwave-assisted synthesis strategy for MOF fabrication facilitated a simplified, more efficient process that resulted in high-quality MOF materials, La-MOF and Zr-MOF. The resulting La-MOF and Zr-MOF materials exhibit impressive surface areas of 76.90 and 293.50 m2/g, respectively, with enhanced thermal stability observed in La-MOF up to 470 °C and Zr-MOF up to 450 °C. The maximum experimental PFOA adsorption capacities of 310 mg/g for La-MOF and 290 mg/g for Zr-MOF highlight their effectiveness in removing PFOA from contaminated water. Our findings further suggest that both La- and Zr-MOF follow pseudo-2nd-order kinetic model and the Langmuir isotherm for PFOA removal, while maintaining stable adsorption efficiency for up to seven cycles. The real-world application of these materials was also investigated revealing that in fixed-bed column tests, breakthrough positions of 150 and 174 minutes were observed for Zr- and La-MOF, respectively, with corresponding bed volumes of 452 mL and 522 mL based on the PFOA limit of 0.07 µg/L. Eventually, our approach not only circumvented the conventional PET recycling practices, but also facilitated the production of high-quality MOFs that demonstrated excellent performance in removing PFOA from water. The synthesis of MOFs from waste PET bottles via a one-pot microwave-assisted approach has the potential to pave the way for the sustainable and efficient treatment of organic pollutants in water environments.”

Reviewer 2 Report

Title: Transforming Waste into Value: Eco-Friendly Synthesis of 2 MOFs for Sustainable PFOA Remediation

The manuscript with title: “Transforming Waste into Value: Eco-Friendly Synthesis of 2 MOFs for Sustainable PFOA Remediation, demonstrates a good potential in the field of Metal Organic Framework MOF-based catalyst for perfluorooctanoic acid (PFOA) water contaminant removal. I find the manuscript convincing and suitable for publication in the Sustainability Journal after minor revision.

The manuscript “Transforming Waste into Value: Eco-Friendly Synthesis of 2 MOFs for Sustainable PFOA Remediation”, demonstrate high performed MOF-based catalysts for the PFOA contaminant removal obtained by reutilization of PET-bottles and using a facile method of synthesis. The materials developed in this study, were well characterised via fundamental methods such as XPS, TGA, FTIR, that are crucial for the study of surface chemistry and structural of carbon materials. In the end the materials were applied for the study of water contaminant removal and application, demonstrating that Zr-MOF and La-MOF performance are comparable with the literature. Therefore, an addition of some corrections along results could significantly improve the scientific impact of the paper and must be performed for the publication.

The following areas should be addressed:

Line 128: Please correct the Celsius degree unity.

A schematic of the PFOA removal process in section 2.5 or supplementary information might be interesting to complement the experimental section. 

Figure 3. Comparing the bending energies in figure 3A with the high-resolution components C1s, O1S and La3d in figures 3B – 3D, respectively, there are shift of the binding energies. Are the respective BE corrected by C1s?

 X-ray photoelectron spectroscopy is one of the most relevant methods to analyse carbon materials/ functionalised carbon materials. I detailed data analysis can complement the understand of the MOF functionality toward the PFOA removal. Please report a table containing each component with respective binding energy, atomic percentage and ratio of oxygen/carbon and metal (Zr and La)/carbon.

A sequence of graphs in figure 3 and 4 needs to be addressed.

In general, the colour of both catalysts Zr-MOF and La-MOF in the figures change for each type of analysis, it sometimes difficult to understand the sample in the graph. I suggest adopting a colour to each sample.

The authors have demonstrated that perfluorooctanoic acid removal is similar for in the presence of both materials La-MOF and Zr-MOF, with 364 and 340 mg/g respectively. However, La-MOF demonstrated more activity toward the contaminant removal. The La-MOF and Zr-MOF structural and surface chemistry characterization demonstrated a highest surface area for the Zr-MOF and lamellar and intercalated framework structure, where La-MOF represents a well-ordered crystalline structure. In addition, thermogravimetric analysis demonstrates a high stability of Zr-MOF with respect to La-MOF.  According to the MOF composition and surface chemistry, witch influence for a higher activity of La-MOF? I propose to the authors introduce a paragraph explaining if the active sites from the metals or oxygen surface groups, higher BET surface area and porosity, can achieve a pivotal role in the final activity of the MOF to the PFOA contaminant removal. Based on XPS complemented by BET, TGA and FTIR and comparing with materials reported in the literature, this addition could enrich the present manuscript.

Author Response

Response to reviewer #2

Reviewer 2 comments and suggestions for authors:

Title: Transforming Waste into Value: Eco-Friendly Synthesis of 2 MOFs for Sustainable PFOA Remediation

The manuscript with title: “Transforming Waste into Value: Eco-Friendly Synthesis of 2 MOFs for Sustainable PFOA Remediation”, demonstrates a good potential in the field of Metal Organic Framework MOF-based catalyst for perfluorooctanoic acid (PFOA) water contaminant removal. I find the manuscript convincing and suitable for publication in the Sustainability Journal after minor revision.

The manuscript “Transforming Waste into Value: Eco-Friendly Synthesis of 2 MOFs for Sustainable PFOA Remediation”, demonstrate high performed MOF-based catalysts for the PFOA contaminant removal obtained by reutilization of PET-bottles and using a facile method of synthesis. The materials developed in this study, were well characterised via fundamental methods such as XPS, TGA, FTIR, that are crucial for the study of surface chemistry and structural of carbon materials. In the end the materials were applied for the study of water contaminant removal and application, demonstrating that Zr-MOF and La-MOF performance are comparable with the literature. Therefore, an addition of some corrections along results could significantly improve the scientific impact of the paper and must be performed for the publication.

The following areas should be addressed:

Comment 1: Line 128: Please correct the Celsius degree unity.

Authors' response to comments 1: Thank you for your comment. It was corrected.

Please see line 149.

Comment 2: A schematic of the PFOA removal process.

Authors' response to comments 2: Thank you for your comment. A schematic illustration was added as Fig.1 according to your suggestion.

Comment 3: Figure 3. Comparing the binding energies.

Authors' response to comments 3: Thank you for your comment. When preparing graphics and coloring for Fig.3, the data was mistakenly entered before correction by C1S. The figure and the respective binding energies were corrected.

Comment 4: X-ray photoelectron spectroscopy is one of the most.

Authors' response to comments 4: Thank you for your comment. Table 1 was added reporting binding energies of components, atomic percentages and ratios as you suggested.

Please see line 359.

Comment 5: A sequence of graphs in figure 3 .

Authors' response to comments 5: Thank you for your comment. It is done.

Comment 6: In general the color of both catalysts.

Authors' response to comments 6: Thank you for your comment. Figures were revised according to your suggestion. 

Comment 7: The authors have demonstrated that.

Authors' response to comments 7: Thank you for your comment. According to your suggestion, a text was added to XPS analysis explaining key role players in PFOA removal process. 

Please see lines 328 to 353.

The added text is as below:

“The corresponding binding energies, atomic percentages, and oxygen to carbon ratio as well as metal to carbon ratios are given in Table 1. The metal-to-ligand ratio and synthesis conditions were identical for both MOFs. However, the composition analysis presented in Table X reveals that the La-MOF exhibits a higher percentage of metals compared to the Zr-MOF. It is reasonable to infer that a larger number of La metal ions in the La-MOF form coordination bonds with the ligands, leading to a higher concentration of La-ligand complexes within the La-MOF structure. Conversely, the lower percentage of Zr in the Zr-MOF indicates a relatively smaller number of Zr metal ions in contact with the ligands. This implies that certain ligands in the Zr-MOF may not be coordinated with Zr metal ions, resulting in the presence of uncoordinated, partially coordinated or free ligands within its structure. The presence of these free ligands in the Zr-MOF could be attributed to factors such as incomplete coordination during synthesis or the presence of structural defects. Consequently, the repulsion between the uncoordinated ligands and PFOA can occur due to electrostatic forces or steric hindrance, which can account for the reduced adsorption capacity of the Zr-MOF compared to the La-MOF. Electrostatic repulsion arises from the repulsive forces experienced between the negatively charged carboxylic groups on both the ligands and PFOA. Steric hindrance occurs when the spatial arrangement of the ligands restricts the penetration of PFOA molecules into the MOF structure, limiting their entrapment. Furthermore, the XRD analysis confirms the aforementioned findings by indicating a lamellar structure for the Zr-MOF, while  the La-MOF exhibits a well-ordered structure. This suggests that certain regions of the Zr-MOF may possess a bulk-like form, possibly due to the presence of mesopores or interlayer spaces between the layers. Despite its less ordered structure, the Zr-MOF retains a significant surface area attributed to these porous regions. The larger surface area of the Zr-MOF further supports the notion that repulsive forces among uncoordinated ligands and PFOA contribute to its reduced adsorption capacity compared to the La-MOF.”

Round 2

Reviewer 1 Report

The revision manuscript has well addressed my initial comments. I would suggest accepting this manuscript after revising following comments:

1. Please recheck the typo for whole documents, especially the new added contents.

2. About label for axis in Fig 8 and 9, although "log" can be written in full name as "logarithm of", I agree that it will be a bit tricky to name others. However, I think log and exponential function will 'clear' the units of parameters so log X should not have any unit. Please confirm and remove the unit after all log X.

3. Line 302 - 305: it is quite awkward to cite only 1 single paper that actually made a better MOF than the ones developed in this manuscript. It will raise a question to readers that why authors didn't use Abid et al.'s method.
I would recommend the authors either providing one statement about why there is a difference, or finding more reference and citing them as a range of surface area xx-yy m2/g.

Author Response

Response to reviewer

Reviewer comments and suggestions for authors:

The revision manuscript has well addressed my initial comments. I would suggest accepting this manuscript after revising following comments:

Comment 1: Please recheck the typo for whole documents, especially the new added contents.

Authors' response to comments 1: Thank you for your comment. The typo was checked for the whole manuscript and mistakes were corrected.

Comment 2: About label for axis in Fig 8 and 9, although "log" can be written in full name as "logarithm of", …

Authors' response to comments 2: Thank you for your comment. It was corrected as you suggested.

Please see lines 409 and 412.

Comment 3:  Line 302 - 305: it is quite awkward to cite only 1 single paper that …

Authors' response to comments 3: Thank you for your comment. The use of PET waste material without purifying the ligand source in a one-pot synthesis strategy in this research results in reduced surface area for MOFs. Abid et al.’s method was an example to show that other researchers who used pure chemicals obtained a larger surface area than our study. However, according to your suggestion some other references were added to the text.  

Please see lines 305 and 308.

The added text is as below:

“Safinejad et al. synthesized an La-MOF using pure chemicals for drug delivery applications with a surface area of 521 m2/g. Usually the surface area obtained for Zr-MOF and La-MOF synthesized from pure chemicals are in the range of 790-2700 m2/g and 240-530 m2/g, respectively.
